# Synthesis of BiOI/Mordenite Composites for Photocatalytic Treatment of Organic Pollutants Present in Agro-Industrial Wastewater

**DOI:** 10.3390/nano12071161

**Published:** 2022-03-31

**Authors:** Alejandra Gallegos-Alcaíno, Nathaly Robles-Araya, Camila Avalos, Alexander Alfonso-Alvarez, Carlos A. Rodríguez, Héctor Valdés, Norma A. Sánchez-Flores, Juan C. Durán-Alvarez, Monserrat Bizarro, Francisco J. Romero-Salguero, Adriana C. Mera

**Affiliations:** 1Departamento de Ingeniería Mecánica, Facultad de Ingeniería, Universidad de La Serena, Benavente 980, La Serena 1700000, Chile; alejandra.gallegosa@userena.cl (A.G.-A.); alexander.alfonso@userena.cl (A.A.-A.); 2Instituto de Investigación Multidisciplinario en Ciencia y Tecnología, Universidad de La Serena, Raúl Bitrán 1305, La Serena 1700000, Chile; arodriguez@userena.cl; 3Departamento de Química, Laboratorio Central de Análisis Químico, Universidad de La Serena, Juan Cisternas 1015, La Serena 1700000, Chile; nathyrobles.araya@gmail.com (N.R.-A.); c.avalosb1303@gmail.com (C.A.); 4Clean Technologies Laboratory, Engineering Faculty, Universidad Católica de la Santísima Concepción, Concepción 4030000, Chile; hvaldes@ucsc.cl; 5Instituto de Ciencias Aplicadas y Tecnología, Universidad Nacional Autónoma de México, Circuito Exterior S/N, Ciudad Universitaria, Ciudad de Mexico 04510, Mexico; norma.sanchez@icat.unam.mx (N.A.S.-F.); carlos.duran@icat.unam.mx (J.C.D.-A.); 6Instituto de Investigaciones en Materiales, Universidad Nacional Autónoma de México, Circuito Exterior S/N, Ciudad Universitaria, Ciudad de Mexico 04510, Mexico; monserrat@materiales.unam.mx; 7Departamento de Química Orgánica, Instituto Universitario de Investigación en Química Fina y Nanoquímica, Facultad de Ciencias, Universidad de Córdoba, 14071 Córdoba, Spain; qo2rosaf@uco.es

**Keywords:** bismuth oxyiodide (BiOI), heterogeneous photocatalytic process, synthetic zeolite, surface response methodology

## Abstract

Recently, bismuth oxyiodide (BiOI) is an attractive semiconductor to use in heterogeneous photocatalysis processes. Unfortunately, BiOI individually shows limited photocatalytic efficiency, instability, and a quick recombination of electron/holes. Considering the practical application of this semiconductor, some studies show that synthetic zeolites provide good support for this photocatalyst. This support material permits a better photocatalytic efficiency because it prevents the quick recombination of photogenerated pairs. However, the optimal conditions (time and temperature) to obtain composites (BiOI/ synthetic zeolite) with high photocatalytic efficiency using a coprecipitation-solvothermal growth method have not yet been reported. In this study, a response surface methodology (RSM) based on a central composite design (CCD) was applied to optimize the synthesis conditions of BiOI/mordenite composites. For this purpose, eleven BiOI/mordenite composites were synthesized using a combined coprecipitation-solvothermal method under different time and temperature conditions. The photocatalytic activities of the synthesized composites were evaluated after 20 min of photocatalytic oxidation of caffeic acid, a typical organic pollutant found in agro-industrial wastewater. Moreover, BiOI/mordenite composites with the highest and lowest photocatalytic activity were physically and chemically characterized using nitrogen adsorption isotherms, scanning electron microscopy (SEM), X-ray diffraction (XRD), Fourier transform infrared spectroscopy (FT-IR), and diffuse reflectance spectroscopy (DRS). The optimal synthesis conditions prove to be 187 °C and 9 h. In addition, the changes applied to the experimental conditions led to surface property modifications that influenced the photocatalytic degradation efficiency of the BiOI/mordenite composite toward caffeic acid photodegradation.

## 1. Introduction

Wastewater from the agro-industrial sector, such as wine, pisco, and olive oil industries, contains a high non-biodegradable organic load with high phytotoxicity, and antimicrobial properties [1,2], largely attributable to the presence of phenolic compounds [3]. It has been demonstrated that phenolic compounds such as gallic acid, caffeic acid, and gentisic acid are toxic to aquatic ecosystems [4,5]. Among the above-mentioned pollutants, caffeic acid is one of the most refractory phenolic compounds, due to its high toxicity and antibacterial activity [6]. Thus, its presence in biological treatment systems causes inhibitory effects or even the death of microorganisms. It has also been proven that it represents a human health risk when it is exposed in small concentrations [6,7].

The use of advanced oxidation processes (AOPs) has become an alternative treatment with great potential due to their high oxidizing capacity in the decontamination of wastewater with organic compounds that are difficult to biodegrade [8,9,10]. Among the AOPs, the heterogeneous photocatalytic processes appear as the most efficient and sustainable treatments for the removal of non-biodegradable pollutants [11]. Recalcitrant compounds are quickly eliminated from the water using solar radiation and at low cost [12,13,14,15]. Furthermore, BiOI is activated under natural sunlight, reducing the costs and improving energy efficiency during treatment [16].

TiO_2_ P-25 is the most studied material in heterogeneous photocatalysis for wastewater treatment because it is a stable and low-cost material. However, the main disadvantages are its high recombination rate, its nanometric size, and its activation requires radiation with a wavelength lower than 390 nm, which corresponds to only 5% of the solar spectrum [12,17]. These disadvantages increase the operating costs and hinder the sustainability of this technology on an industrial scale [11].

Considering the above limitations of this semiconductor, current studies have focused on the synthesis of photocatalysts with great potential such as bismuth oxyiodide (BiOI) due to its proven efficiency in the degradation and mineralization of different non-biodegradable or low-biodegradable organic pollutants [18,19]. Some studies show that the photocatalytic activity of BiOI is higher than TiO_2_ P-25 under simulated solar irradiation [12,18]. However, the nanometric size of BiOI leads to the same disadvantages that the rest of powder photocatalysts. Thus, researchers have focused mainly on the synthesis of photocatalysts that are activated with visible light, with high photocatalytic efficiency, low recombination rate, and that can be reused easily [20].

Unfortunately, BiOI individually shows limited photocatalytic efficiency and quick recombination of photogenerated charge carriers. In addition, BiOI photocatalyst presents instability and a difficult recovery [21]. Recently, several studies have shown the design and synthesis of some composite materials to overcome these disadvantages using inert porous supporters to immobilize these nanometric materials [22], in order to prevent the quick recombination of electron/holes.

The supports used include zeolites, mosquito nets, polymers, glass, activated carbon, graphene, silicon oxide, ceramic fibers, magnetic material, and cellulose [23,24,25,26,27,28,29,30,31,32,33]. Studies have reported that zeolites are good support materials because they exhibit high thermal stability, high photostability, they are ion exchangers, and can adsorb substances selectively depending on their molecular size [30]. The synthetic mordenite-type zeolite is one of the most important because it has a high ratio of silicon to aluminum atoms. This characteristic makes mordenite more resistant to acids than other zeolites [34,35]. In addition, other authors have reported that semiconductor nanoparticles (e.g., BiOI) grow strongly adhered on the mordenite surface. This generates several technological advantages, such as preventing the detachment of the semiconductor during the photocatalytic reaction and increasing the stability of the new composite [36].

Therefore, by combining the properties of a synthetic mordenite-type zeolite with the properties of a BiOI semiconductor, the new BiOI/mordenite composite can increase the photocatalytic efficiency of the process because it decreases the electron/hole recombination rate [26,30,37]. In addition, obtaining BiOI/mordenite composite can have a potential application for wastewater treatment facilitating the degree of reuse of the photocatalyst.

On this matter, it has been shown that it is possible to couple BiOI to zeolite, keeping its high photocatalytic activity. For example, Zhao et al. [21] synthesized a three-dimensional BiOI/synthetic zeolite composite by a solvothermal route at 180 °C for 7 h. The synthesized material achieved around a 95% of photocatalytic activity in the oxidation of methylene blue in water after 10 min of visible radiation. However, to our knowledge, there are no data on BiOI supported on mordenite; moreover, there is a lack of information related to the effect of different experimental conditions (e.g., temperature and time) on the physical-chemical properties of BiOI/synthetic zeolite composites, as well as their impact on the photocatalytic efficiency. In addition to the previously mentioned, optimal synthesis conditions such as time and temperature to obtain BiOI/synthetic zeolite composites with a high photocatalytic activity using a combined coprecipitation-solvothermal growth method have not been reported yet.

This work aims to determine the optimal temperature and time for the synthesis of BiOI/mordenite composite that leads to the highest photocatalytic degradation of caffeic acid. Composites were obtained by the coprecipitation-solvothermal growth route, temperature ranges from 126 °C to 180 °C, and time ranges from 7 to 18 h. For this purpose, a CCD factorial design and a response surface methodology (RSM) were used [38,39]. In addition, the composites with the lowest and highest photocatalytic activity, as well as the material obtained under optimal conditions were characterized. It was determined that the changes applied in the experimental conditions led to surface properties modifications that influenced the photocatalytic efficiency of the BiOI/mordenite composites.

## 2. Materials and Methods

### 2.1. Materials

All the reagents used in the synthesis of BiOI/mordenite composites were of analytical grade. For this purpose, bismuth nitrate pentahydrate Bi(NO_3_)_3_∙5H_2_O (Sigma-Aldrich, Toluca, México, 99.0%) was used as Bi^3+^ ion source; potassium iodide KI (Merck, Darmstadt, Germany, 99.0%) was employed as a source of I^–^ ions, and a mixture of absolute ethanol (Merck, Darmstadt, Germany, 99.5% *v*/*v*) and deionized water was used as a solvent. A synthetic mordenite-type zeolite in its hydrogen form with a SiO_2_/Al_2_O_3_ ratio of 90 and surface area of 500 m^2^ g^−1^ (coded as CBV 90A) (Zeolyst Int. Farmsum, The Netherlands). was applied in the synthesis of the composite. A solution of caffeic acid (10 mg L^−1^) was used as the target organic contaminant, representative of phenolic pollutants normally found in agro-industrial wastewaters.

### 2.2. Experimental Design and Statistical Analysis

A design of experiments (DoE) based on multivariate analysis was applied through a central composite design (CCD) that comprised a two-level factorial design with three center points and four-star points [38,40,41] to obtain the BiOI/mordenite composite under optimum synthesis conditions by the solvothermal coprecipitation/growth method. The variables were coded using unit values, where −1 and +1 were defined as the lowest and highest value, respectively. The center point was coded as 0 and determined in triplicate. The influence of temperature and reaction time during the synthesis of the BiOI/mordenite composites was evaluated after 20 min of photocatalytic oxidation of caffeic acid under simulated solar radiation. Thus, the removal efficiency (%) was taken as the response variable (Y). The minimum and maximum values of temperature were fixed at 126 °C and 180 °C, respectively; whereas the minimum and maximum values of time were established at 7 and 18 h, respectively, according to the values reported in the literature [18,21]. The number of experimental runs (N) was obtained using the CCD model, as follows:N = 2^k^ + 2K + n_c_(1)
where K stands for the number of natural variables, 2^K^ represents the factorial points (maximum and minimum of each variable), 2K is the axial or star points (new maximum and minimum extremes of each factor), and n_c_ represents the number of central points [42]. According to this DoE, eleven experimental runs were determined and organized in random order, as shown in Table 1.

The reaction over the first 20 min was assumed to follow a pseudo-first-order kinetic model as expressed in Equation (2), which is generally used in photocatalytic oxidation processes of organic pollutants in water when the concentration of the pollutant is low [43].
(2)$$\ln\frac{C_{0}}{C}=kt$$
where $C_{0}$ and C are the concentration of the target pollutant in the solution at time 0 and t, respectively, and k is the pseudo-first-order rate constant (s^−1^) [44,45].

The standardization of the experimental conditions to test the influence of the variables of temperature and reaction time on the synthesis of BiOI/mordenite was validated by the ANOVA statistical test with a 95% confidence level [39]. The determination of the polynomial that describes the influence of the variables and the response surfaces was performed using the commercial software MODDE PRO 12.0.1.

### 2.3. Synthesis of BiOI/Mordenite Composites

The synthesis of BiOI/mordenite composites was developed by a combined coprecipitation-solvothermal growth method using the route shown in Figure 1. For this purpose, 10 mL of absolute ethanol was added to 0.1638 g of Bi(NO_3_)_3_∙5H_2_O, leaving the solution under constant stirring (stage 1). Subsequently, the solution was subjected to an ultrasonic bath and then 0.5 g of calcined mordenite was added (stage 2). Simultaneously, 0.056 g of KI was dissolved in 10 mL of deionized H_2_O, left in the ultrasonic bath (stage 3). Then, the KI solution was added dropwise to the Bi(NO_3_)_3_∙5H_2_O solution prepared earlier (stage 4). After the addition was completed, it was stirred constantly for an additional 60 min. Then, 15 mL of the resulting solution was transferred to a 23 mL Teflon-lined autoclave reactor, fixing the selected temperatures and reaction times (stage 5), as established in the DoE shown in Table 1.

At the end of each experimental run, the reactor was cooled to room temperature. The obtained materials were separated in a vacuum filtration system and washed alternately with 50 mL of absolute ethanol and 50 mL of deionized water until 100 mL of each solvent was completed. Once this process was completed, the samples were dried at 80 °C for 12 h in the Carbolite Gero CWF 1100 flask (stage 6). Once the drying process was finished, the obtained composites were stored in labeled amber bottles.

### 2.4. Photocatalytic Activity Assessment of Synthesized BiOI/Mordenite Composites

The photocatalytic activity of the 11 synthesized materials was evaluated following the removal efficiency of caffeic acid after 20 min of reaction time under simulated solar radiation. A xenon lamp (VIPHID 6000 K, 12 W) with a spectral range of 380–900 nm was used for this purpose.

The photocatalytic tests performed with the 11 materials were carried out in a borosilicate batch reactor (Figure 2), which contained 250 mL of caffeic acid solution at a concentration of 10 mg L^−1^ and 0.075 g of the BiOI/mordenite composite, at natural pH and room temperature. In order to reach the adsorption–desorption equilibrium, the photocatalytic system remained in absolute darkness for 40 min before switching on the xenon lamp. After the previous 40 min elapsed, the photocatalytic reaction was activated by the radiation of the xenon lamp for 60 min. Ten samples of 10 mL were taken for the 11 photocatalytic tests at the following times: [−40, 0, 2, 2, 5, 5, 10, 10, 20, 30, 30, 40, 40, 50, 60] minutes of reaction. The samples were filtered through a nitrocellulose membrane (Millipore, 0.22 μm) to separate the composite from the solution and left in the dark. Subsequently, the concentration of caffeic acid was measured by taking the UV-vis absorbance spectra measuring the absorbance between 200 and 500 nm in a spectrometer (UV/Vis Evolution 220 Thermo Scientific) (Madison, WI, USA).

### 2.5. Stability Tests

The stability of the obtained composite BiOI/mordenite under optimal conditions was determined through two consecutive reaction cycles. At the end of the reaction, the solid was separated from the aqueous phase by filtration for 20 min and washed with distilled water to remove ionic and organic residues. Then, the recovered solid was dried at 80 °C for 12 h and weighted for a new reaction cycle.

### 2.6. Physical-Chemical Characterization of Prepared Materials

A physical-chemical characterization was carried out on the following five materials: synthetic mordenite zeolite, the pure BiOI photocatalyst synthesized by the solvothermal method, the BiOI/mordenite composites which showed the lowest and highest photocatalytic activity, and the BiOI/mordenite composite under optimum conditions of synthesis by the solvothermal coprecipitation/growth method.

The morphology of each material was determined by a scanning electron microscope (SEM) operated at an accelerating voltage of 25 kV, using a JEOL T-300 microscope (Tokyo, Japan). Specific surface area and pore size distribution of the materials were determined by nitrogen adsorption–desorption isotherms at 77 K using the Brunauer–Emmett–Teller (BET) analysis and by the Barret–Joyner–Halenda (BJH) method, respectively, on a Micromeritics 3Flex Version 4.02 adsorption analyzer (Norcross, GA, USA). The crystallinity and phases of the synthesized material were identified by X-ray diffraction (XRD) (Ettlingen, Germany), using a Bruker D4 powder diffractometer equipped with a Lynxeye detector and operated with Cu radiation with Ni K beta filter; the recording covered an angular range between 3 and 70° 2θ with a step of 0.020°. The functional and anchoring groups on the surface of the materials were identified by Fourier transform infrared spectroscopy (FT-IR) using a Thermo iS-50 FTIR spectrometer with an MCT detector at a resolution of 1 cm^−1^ and 32 scans. The optical characteristics of the materials were studied by diffuse reflectance spectroscopy (DRS), using a Thermo Scientific Evolution 220 UV-Visible spectrophotometer (Madison, WI, USA) with an integrating sphere.

## 3. Results and Discussion

### 3.1. Optimization of BiOI/Mordenite Composite Synthesis

Eleven BiOI/mordenite composites were obtained by multivariate CCD analysis. The experiments were performed by simultaneously varying the synthesis temperature and reaction time.

In order to obtain the optimal synthesis conditions of the BiOI/mordenite composites, experimental removal efficiencies were compared to the predicted removal efficiencies by the CCD statistical model, after 20 min of photocatalytic reaction. After this reaction time, no significant difference was observed in the photocatalytic efficiency of the 11 synthesized composites.

The results listed in Table 1 indicate that removal efficiencies are significantly variable. Some temperatures and times are more favorable than other conditions to obtain a BiOI/mordenite composite with better photocatalytic efficiency. Among all experiments, runs 4, 5, 6, and 7 show the highest removal efficiencies.

Subsequently, in order to corroborate if the model was statistically valid, an ANOVA test was performed with 95% confidence. The model proposed in this study has a coefficient of determination (R^2^) of 0.993, which determines the level of variation of the response as a function of the variability of the parameters temperature and time, as well as their interactions [46].

The predictive validity of the method was evaluated using the Q^2^ value (0.956), which represents the ability of the model to predict the degradation of pollutants, based on any combination of temperature and time variables in the synthesis of the material that was within the domain used in this study [41].

The polynomial shown in Equation (3) was obtained by fitting the experimental data (*Y*) by multiple linear regression to a polynomial model using the CCD design. The values in parentheses represent the standard deviation of each of the coded variables. The response polynomial allows predicting the response of removal efficiency (*Y*) and represents the importance of the variables, that is, the effect of temperature (*T*) and time (*t*) on the synthesis process of the BiOI/mordenite composite by the applied coprecipitation-solvothermal growth method.
(3)$$Y\;\left(removal\;efficiency\right)=41.23\;\left(\pm 0.82\right)+2.44\;T\left(\pm 0.50\right)-7.56\;t\;\left(\pm 0.50\right)-2.08\;T^{2}\left(\pm 0.60\right)-11.39\;t^{2}\left(\pm 0.60\right)-6.02\;T\cdot t\;\left(\pm 0.70\right)$$

According to Equation (3), it can be observed that there is a complex relationship between temperature and time. It is clear that the reaction time is an important variable in the synthesis of BiOI/mordenite composites, where its negative magnitude (7.56 t) indicates a reduction in the photocatalytic activity as the reaction time increases. On the contrary, increasing the temperature leads to a positive influence on the photocatalytic activity. It should be noted that the augmented activity as the temperature increases is limited by the quadratic expression of temperature in Equation (3). This can be observed in Table 1, where an increase in the temperature resulted in a higher photocatalytic removal efficiency only until a certain value of T (see experimental runs 4, 9, and 10).

The impact of each term of Equation (3) on the photocatalytic removal efficiency is displayed in Figure 3. It is possible to see that the quadratic effect associated with both variables (T^2^ and t^2^) is negative. In addition, there is a negative synergy between the variables of time and temperature in the synthesis process of the BiOI/mordenite composites. Since the temperature and time are varied simultaneously, the photocatalytic activity of the composite decreases significantly for higher values of T and t.

Figure 4 displays a three-dimensional representation of the response polynomial (Equation (3)) for the synthesis of the BiOI/mordenite composite. It can be observed that the time of synthesis is the most important factor influencing the removal efficiency of caffeic acid. The red-colored area in the response surface (Figure 4) corresponds to the region where the synthesized composites have the highest removal efficiency values of the target pollutant. Thus, the best conditions of temperature and time in the synthesis process result to be 187 °C and 9 h, respectively, to obtain the BiOI/mordenite composite with the highest photocatalytic activity under simulated solar radiation. In the optimal conditions of photocatalyst synthesis, the removal efficiencies predicted by the statistical model using MODDE PRO 12.0.1 software are in the range of 48.5–51.5%, in which the highest photocatalytic activity towards caffeic acid elimination is obtained after 20 min of reaction.

In order to validate the range of the predicted removal efficiencies and to corroborate the reliability and reproducibility of the statistical model, a BiOI/mordenite composite sample was synthesized under the optimized conditions (187 °C and 9 h). The photocatalytic activity of the optimized composite was evaluated, and an average value of removal efficiency of 49.9% was obtained from three photocatalytic assays, as shown in Table 2. As it can be noticed, the removal percentage is within the range predicted by the MODDE PRO 12.0.1 software, validating the applied statistical model to obtain the BiOI/mordenite composite with the highest photocatalytic activity for the degradation of caffeic acid in water. It is worth mentioning that the observed removal efficiency of the obtained composite under optimal conditions is higher than the value achieved when pure BiOI was used (42.8%). Such experimental pieces of evidence indicate that BiOI supported on the synthetic zeolite increases the photocatalytic activity.

The photocatalyst stability of the composite BiOI/mordenite sample obtained in optimal conditions was evaluated in two consecutive reaction cycles of 20 min reaction time. Results shown in Figure 5 indicate that the removal efficiency remains almost the same after the two applied operating cycles, with values of 51.42% and 49.22% in the first and second cycle, respectively. Such evidence suggests that the BiOI/mordenite composite obtained under the optimized synthesis conditions could be used as a promising candidate for the photocatalytic oxidation of phenolic compounds present in the wastewater of the agroindustry without losing its photocatalytic activity.

### 3.2. Physical and Optical Properties, and Chemical Surface of the Synthesized Materials

Figure 6 displays the diffraction patterns of pure BiOI (curve a), mordenite zeolite (curve b), and BiOI/mordenite composites with the lower (curve c) and higher (curve d) photocatalytic activities. Moreover, the diffraction pattern of BiOI/mordenite composite obtained at optimal synthesis conditions is also shown (curve e).

The synthesized BiOI (curve a) shows eight characteristic diffraction peaks which are located at 2θ 19.4°, 29.6°, 31.6°, 39.4°, 45.4°, 51.3°, 55.3°, and 66.3°, which are attributed to the (002), (102), (110), (004), (200), (114), (212) and (214) planes, respectively, of the tetragonal structure for BiOI (JCPDS No. 10-0445). Such diffraction peaks confirm the high purity of the synthesized BiOI nanoparticles. The diffraction peaks of mordenite (curve b) are located at 6.5°, 9.77°, 13.45°, 19.61°, 22.40°, and 25.8°, which are ascribed to the (110), (200), (111), (330), (150) and (202) planes, respectively, of the orthorhombic phase of the mordenite structure (JCPDS No. 43-0171). The X-ray diffraction patterns of the BiOI/mordenite composites obtained at different synthesis conditions show not only the typical diffraction peaks of BiOI but also the diffraction peaks of mordenite. However, the material obtained under the optimal synthesis condition shows more intense and defined peaks (curve e). For a better understanding of this higher peak intensity of the sample obtained under optimal conditions, the crystallite size of each composite was calculated using Scherrer’s equation. It was obtained that the crystallite size was almost constant in the range of 37–39 nm without any trend. It means that there is no higher crystallinity in the sample obtained under optimal conditions, and therefore, the observed higher peak intensity could be attributed to more presence of BiOI over the mordenite surface.

The SEM micrographs in Figure 7 display single BiOI (a), single mordenite zeolite (b), and BiOI/mordenite composites synthesized at different time and temperature conditions (c, d, and e).

The SEM micrograph of the single BiOI sample (Figure 7a) exhibits a hierarchical microsphere morphology that is formed by nanosheets with a smooth and irregular surface, with an average size of 4 µm. In the case of the mordenite zeolite (Figure 7b), the SEM micrograph shows structures of different sizes, with an irregular morphology of porous surface and conglomerate zones, indicating that the spatial distribution of the material is not uniform. Moreover, it is observed in the SEM micrograph of all BiOI/mordenite composites that BiOI nanostructures were deposited on the mordenite surface. On the surface of these materials, a morphology similar to three-dimensional flowers is shown, which is composed of interlaced nanosheets with a smooth-bidimensional surface. Such evidence suggests that the mordenite zeolite favors the formation of 3D nanoflower-like BiOI morphologies. These results are consistent with those obtained by Zhao et al. [21]. In particular, the BiOI/mordenite composite which presented the lowest photocatalytic activity (synthesized at 153 °C and 20.3 h) on caffeic acid removal, displays few 3D-flower BiOI nanostructures and two-dimensional thin nanosheets with irregular smooth layer structure (Figure 7c). These structures are isolated over the mordenite surface. In contrast, the BiOI/mordenite composite showing the highest photocatalytic activity (synthesized at 180 °C and 7 h) exhibits the growth of BiOI microspheres over the zeolite surface, with an average diameter of 3.5 µm (Figure 7d). At the same time, the material obtained under optimal synthesis conditions (187 °C and 9 h) presents a large number of two-dimensional thick layers, whose nanostructures are similar to a 3D flower, positioned over the mordenite surface (Figure 7e). In addition, it presents a greater area of exposed surface than the materials synthesized with other conditions of time and temperature (see Table 3). Such experimental pieces of evidence indicate that the experimental synthesis conditions influence the morphology of the synthesized composite materials. The growth of 3D nanoflower-like BiOI structures over the surface of mordenite is favored by the synergy generated between time and temperature in the ranges of synthesis conditions used in this study. These results are congruent with those reported in the literature [12,47,48,49]. It has been indicated that the time variable is a key parameter in the synthesis procedure and the removal rate of the target organic compound used in this study. Shorter synthesis time favors the formation of nanostructures (3D flower) with high photocatalytic activity [18,47,50]. Likewise, the temperature of the synthesis procedure has a positive effect up to a tipping point. In this study, the highest temperature value (187 °C) favored the formation of thick BiOI plates. These nanosheets have a larger exposed area on the amorphous mordenite surface, which increased the photocatalytic activity of the composite. This is consistent with the findings reported by other authors [12,47,51,52].

Table 3 shows the values of the surface and optical properties of the analyzed samples. BET surface analysis shows that BiOI/mordenite composites have a higher specific surface area compared to the BiOI sample, but lower to the mordenite zeolite sample. Moreover, the nitrogen adsorption–desorption data of the optimized sample shows an isotherm of type IV with an H3 hysteresis loop, which is characteristic of mesoporous materials with high adsorption energy [49,53]. The optimized composite sample (187 °C and 9 h) exhibits the highest specific surface area with an average pore size distribution of 3.0 nm, which could be linked to the high observed value in the photocatalytic activity. In composite samples, mordenite not only contributes to a bigger surface area where BiOI microspheres are anchored, but also provides better dispersion of BiOI microspheres over the whole mordenite surface, enhancing the photocatalytic activity of the semiconductor. It has been pointed out in several studies that the surface area is an important factor in the generation of more active sites, improving the photocatalytic activity [53,54,55].

Figure 8 depicts the infrared spectra of BiOI/mordenite composites with the lowest (153 °C/20.3 h) and the highest (180 °C/7 h) photocatalytic activity, as well as the composite synthesized under optimal conditions (187 °C/9 h).

As can be seen, all the FT-IR spectra show characteristic IR bands of the tetragonal crystalline bonds of BiOI around 770 and 1373 cm^−1^, which are originated from asymmetric stretching vibrations of the Bi-O bonds [18,56,57]. These signals are more intense in the composite obtained under optimal conditions, confirming a higher amount of BiOI over this composite sample. In addition, typical bands associated with mordenite zeolite are located at 808, 960 cm^−1^ and in the range between 1206 and 1869 cm^−1^, corresponding to the fundamental vibrations of the tetrahedral bonds of (Al, Si) O_4_ [58,59,60]. Furthermore, stretching signal vibrations associated with silanol groups (Si-O-H) are registered around 3668 cm^−1^ [61]. All samples exhibit two IR bands that can be attributed to hydroxyl groups. The signal around 3500 cm^−1^ corresponds to symmetric stretching vibrations of -OH groups and adsorbed water on the surface of the materials analyzed [62,63]. It is possible to observe that the material with the lower photocatalytic activity (curve a) shows a band with less amplitude in this region than the material with the highest photocatalytic activity (curve b), and the material obtained under optimal conditions (curve c). Similar behavior is observed in the signal attributed to the presence of -OH groups around 1600 cm^−1^ [56,57,64,65]. Such results evidence that the composites with the highest photocatalytic activity and the one obtained under optimal conditions present a higher surface density of -OH groups. The presence of -OH groups seems to play an important role in the photocatalytic activity of the synthesized composites. Hydroxyl groups may act as anchor groups where organic pollutants could be adsorbed [39].

The optical properties of synthesized materials are listed in Table 3 and are expressed as the band gap energy values (*E_g_*) of the synthesized materials. The values were calculated using the Tauc representation (Equation (4)). This method provides a better approximation than the direct extrapolation of the UV-vis spectrum to find the λ of absorption onset [66]:(4)$$\alpha \;\left(hv\right)=A\;{\left(hv-E_{g}\right)}^{n/2}$$
where *α* represents the absorption coefficient, *A* is a constant, *h* the Planck constant, *v* the radiation frequency, *E_g_* the forbidden band energy of the nanoparticles, and the value of the *n*/2 exponent denotes the nature of the electronic transition.

As can be seen, the single BiOI and the obtained composites present similar band gap values (1.93 eV). Such results indicate that the optical band gap of BiOI/mordenite composites are not affected by the synthesis conditions applied in this study.

### 3.3. Mechanistic Approach

Figure 9 displays a proposal for the mechanism growth of BIOI/mordenite composites for each studied condition. Experimental evidence obtained in this study indicates that BiOI tends to fill the zeolite pores, leading to the observed reduction in the composite’s surface area as compared to a single mordenite sample (see Table 3). In addition, the composite obtained at 187 °C and 9 h exhibited a higher amount of BiOI structures anchored over the zeolite surface compared to the samples obtained at 153 °C and 20.3 h and 180 °C and 7 h.

Such results are confirmed by the XRD, SEM, FTIR, and BET analysis. Similarly, the composite obtained under optimal conditions (187 °C and 9 h) exhibited the highest surface area among the three BiOI/mordenite composites. This could be associated with the higher amount of BiOI deposited on the zeolite surface. This suggested growth mechanism of BiOI on the zeolite surface is supported by SEM images, the increasing peak intensity in XRD patterns, signals more amplitude and intensity in FTIR, and higher BET. The highest photocatalytic removal efficiency towards the selected target pollutant obtained by the BiOI/mordenite composite, synthesized under optimal conditions in comparison to the samples with minor photocatalytic efficiency, comes as a result of the physical-chemical surface modification of the composite. In the optimized BiOI/mordenite composite, a synergetic mechanism occurs. Mordenite does not only contribute with a surface where BiOI microspheres are anchored but also to the adsorption of caffeic acid molecules at Brønsted acid sites. Thus, the presence of a higher amount of BiOI structures over mordenite surface act as semiconductors, activated by simulated solar radiation and generating pairs of electrons and holes that interact with water and dissolved oxygen, generating hydroxyl and superoxide radicals that react with caffeic acid molecules adsorbed in the vicinity, and then radical reactions proceed in the bulk-liquid phase, leading to the observed removal efficiencies. In addition, the presence of zeolite in composites favors better stability, slow recombination of electron/holes and enhances a practical application of these composites.

## 4. Conclusions

The synthesis procedure to obtain a BiOI/mordenite composite using a combined coprecipitation-solvothermal growth method was successfully optimized. The surface properties of the synthesized composites were strongly determined by the applied synthesis conditions. The implemented surface response methodology based on a central composite design contributed accurately to determining the optimal temperature and time conditions (187 °C and 9 h) to attain a BiOI/mordenite composite with high photocatalytic efficiency in the removal of caffeic acid from water. Moreover, the experimental design and the analyzed properties demonstrate that there is a synergy between the evaluated experimental conditions (temperature and time) that allows a BiOI/mordenite (90) composite to synthesize with high photocatalytic efficiency, establishing a highly reliable and reproducible method to obtain this type of materials. Therefore, the BiOI/mordenite (90) composite obtained in this study under optimal conditions is a material with high potential for use in the photocatalytic degradation of phenolic compounds present in water. The novel BiOI/mordenite composite obtained in this study could be used in the photocatalytic oxidation of phenolic compounds normally present in the wastewater from the agro-industrial sector.

## Figures and Tables

**Figure 1 nanomaterials-12-01161-f001:**
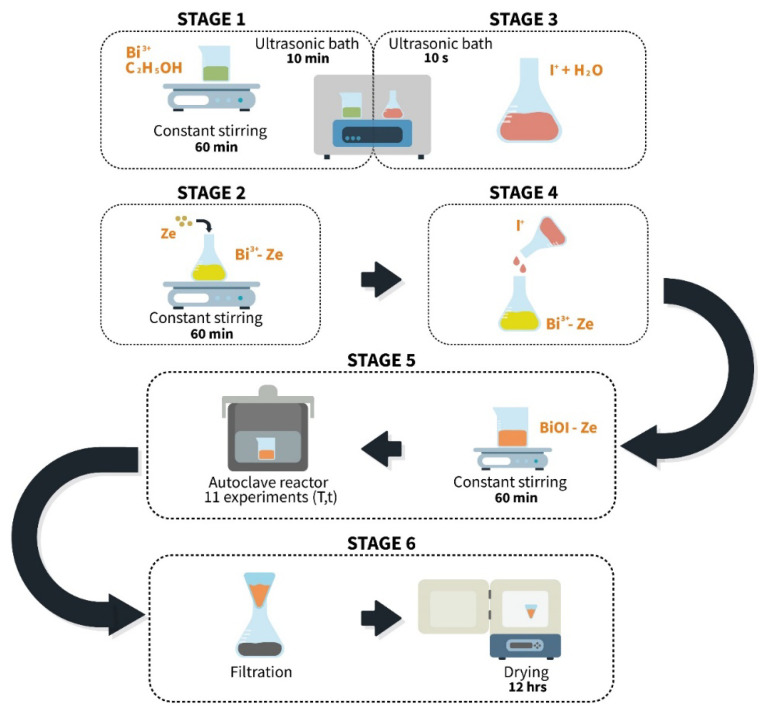
Graphical representation of the synthesis procedure to obtain BiOI/mordenite composites using a combined coprecipitation-solvothermal growth method.

**Figure 2 nanomaterials-12-01161-f002:**
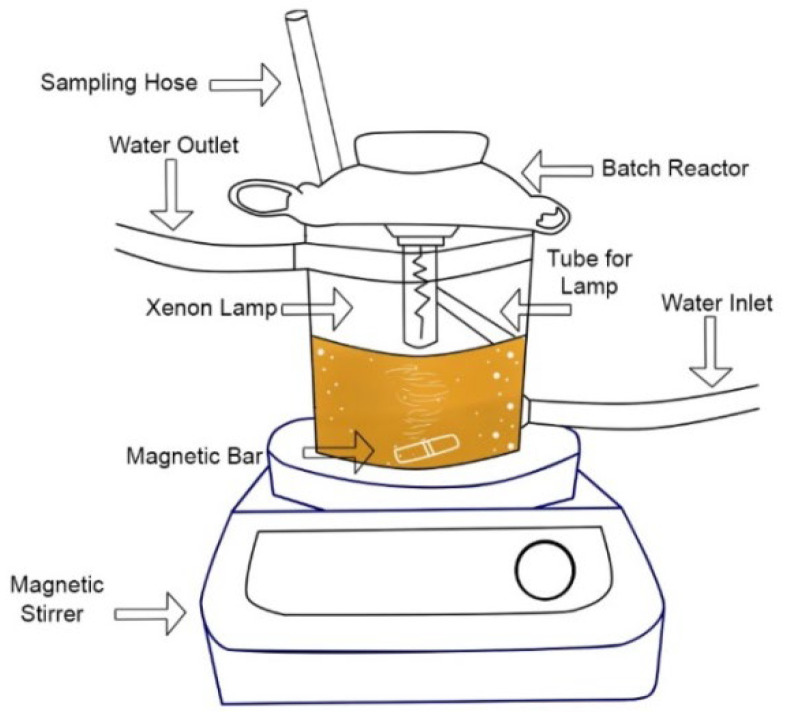
Experimental set-up for the photocatalytic activity assays.

**Figure 3 nanomaterials-12-01161-f003:**
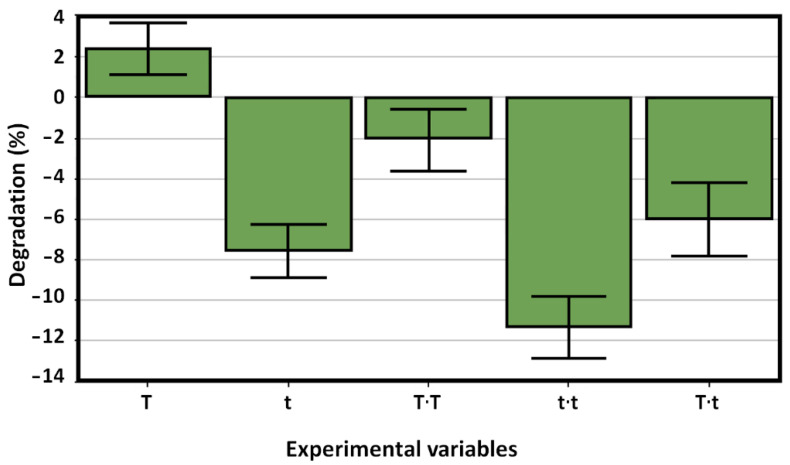
Statistical analysis of the polynomial response of the impact of temperature (T) and time (t) on the photocatalytic removal efficiency.

**Figure 4 nanomaterials-12-01161-f004:**
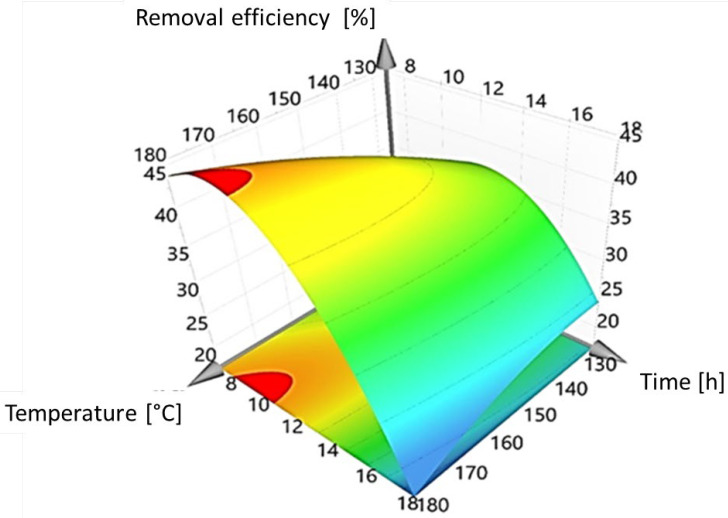
Three-dimensional response surface for the photocatalytic activity of synthesized BiOI/mordenite composites expressed as removal efficiency of caffeic acid under different synthesis conditions of temperature and time.

**Figure 5 nanomaterials-12-01161-f005:**
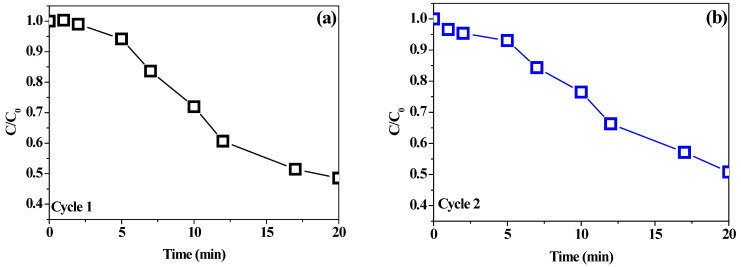
Stability test of optimized BiOI/mordenite composite sample after two operating cycles: (**a**) first cycle, (**b**) second cycle.

**Figure 6 nanomaterials-12-01161-f006:**
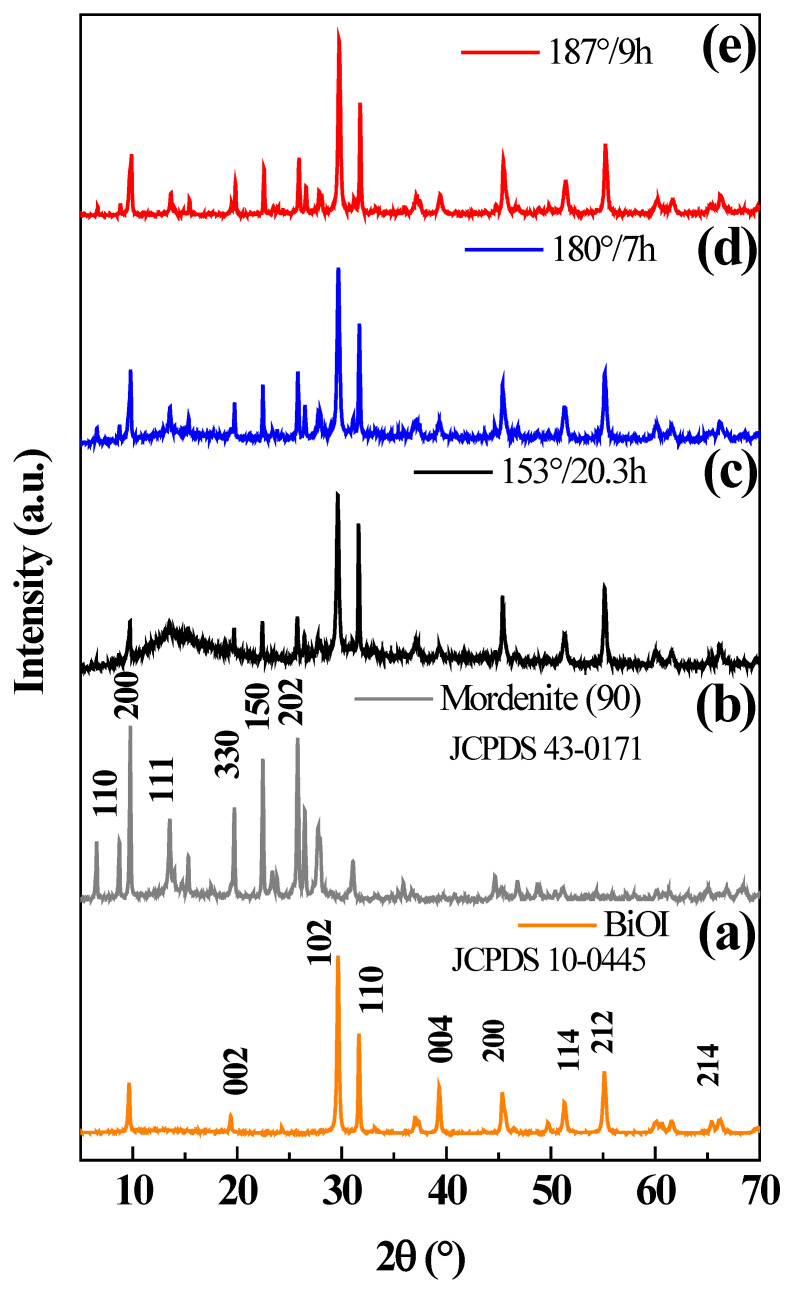
X-ray diffraction patterns of synthesized materials: (**a**) pure BiOI, (**b**) mordenite, (**c**) BiOI/mordenite composite synthesized at 153 °C during 20.3 h, (**d**) BiOI/mordenite composite obtained at 180 °C during 7 h, (**e**) optimized BiOI/mordenite composite synthesized at 187 °C during 9 h.

**Figure 7 nanomaterials-12-01161-f007:**
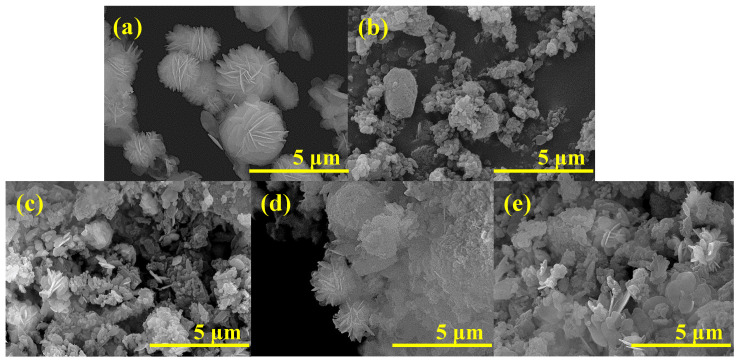
SEM images: (**a**) BiOI, (**b**) mordenite, (**c**) BiOI/mordenite composite synthesized at 153 °C during 20.3 h, (**d**) BiOI/mordenite composite obtained at 180 °C during 7 h, (**e**) optimized BiOI/mordenite composite synthesized at 187 °C during 9 h.

**Figure 8 nanomaterials-12-01161-f008:**
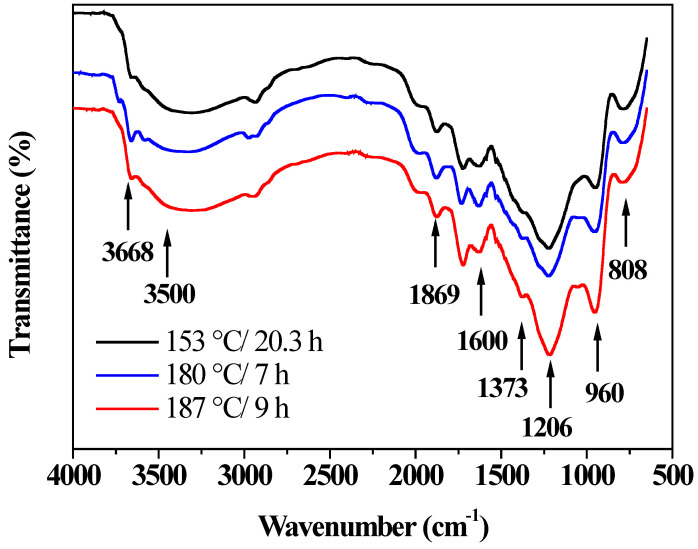
FTIR spectra of BiOI/mordenite composites synthesized at different conditions of temperature and time.

**Figure 9 nanomaterials-12-01161-f009:**
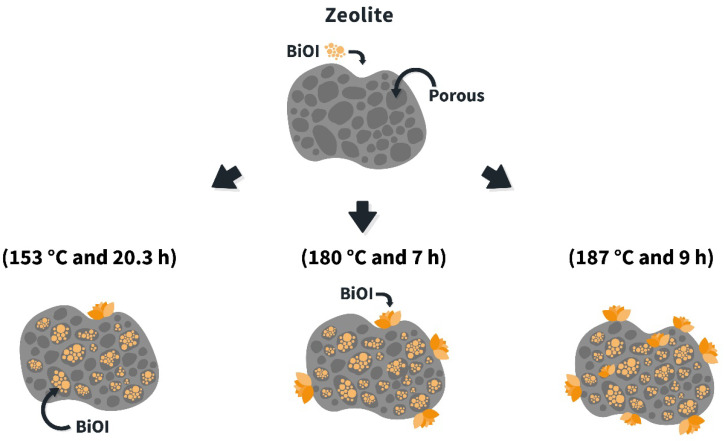
Schematic representations of BiOI/mordenite composite formation.

**Table 1 nanomaterials-12-01161-t001:** Experimental design matrix for the synthesis of BiOI/mordenite composites and response values expressed as removal efficiencies (Y, %) obtained after 20 min of photocatalytic oxidation of caffeic acid under simulated solar radiation.

Experimental Runs	Natural Variables		Coded Variables		Response Values	
	Temperature (°C)	Time (h)	X_1_	X_2_	Y Experimental (%)	Y Predicted (%)
1	153.0	20.3	0	$+\sqrt{2}$	6.1	7.6
2	180.0	18.0	+1	+1	17.8	16.6
3	153.0	4.7	0	$-\sqrt{2}$	29.4	29.0
4	153.0	12.5	0	0	42.2	41.2
5	153.0	12.5	0	0	41.2	41.2
6	180.0	7.0	+1	−1	43.6	43.8
7	153.0	12.5	0	0	40.3	41.2
8	126.0	7.0	−1	−1	27.1	26.9
9	191.2	12.5	$+\sqrt{2}$	0	40.1	40.5
10	114.8	12.5	$-\sqrt{2}$	0	32.6	33.6
11	126.0	18.0	−1	+1	25.4	23.8

X_1_ coded temperature variable, X_2_ coded time variable.

**Table 2 nanomaterials-12-01161-t002:** Comparison of the photocatalytic activities of the BiOI/mordenite composite synthesized under the optimal experimental conditions (187 °C during 9 h) with pure BiOI.

Photocatalytic Tests	Experimental Removal Efficiency (%)	Predicted Removal Efficiency Ranges (%)
1	50.3	48.5–51.5
2	47.9	48.5–51.5
3	51.5	48.5–51.5
Average	49.9	
Individual BiOI	42.8	

**Table 3 nanomaterials-12-01161-t003:** Textural and optical properties of parent materials and BiOI/mordenite composites synthesized at different conditions of temperature and reaction time.

Materials	BET (m^2^ g^−1^)	Pore Diameter (nm)	Pore Volume (cm^3^ g^−1^)	Eg (eV)
Pure BiOI	7	16.3	0.031	1.93
Mordenite	547	2.40	0.331	---
BiOI/mordenite: 153 °C/20.3 h	318	3.18	0.255	1.93
BiOI/mordenite: 180 °C/7 h	360	2.61	0.238	1.93
BiOI/mordenite: 187 °C/9 h	371	3.01	0.282	1.93

## Data Availability

Not applicable.

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
