# Peer review of "Synthesis of BiOI/Mordenite Composites for Photocatalytic Treatment of Organic Pollutants Present in Agro-Industrial Wastewater"

_nanomaterials, 2022, doi:10.3390/nano12071161_

Round 1

Reviewer 1 Report

This paper describes the synthesis of BIOI/Mordenite composites and their application to the photocatalytic treatment of organic pollutants present in agro-industrial wastewater. I accept this paper for publication after minor revision based on the following reasons.

  1. The paper is written well.
  2. All the methods and characterization results are very well explained.
  3. The conclusions and findings are well supported by the results.
  4. The research design is scientifically aligned.

Minor revision requirements are as follows.

  1. English editing is required.
  2. There are typo errors that should be revised.
  3. The authors should emphasize the importance of BIOI/Mordenite in reference to its efficiency and application for wastewater treatment.
  4. The authors must describe in detail why they selected BIOI/Mordenite for their current research.
  5. The authors should add a brief explanation or a table describing the comparison between the efficiency of their synthesized material with others already reported in the literature.
  6. The authors should site some important work related to the degradation of pollutants.

https://doi.org/10.1016/j.jenvman.2021.112605

https://doi.org/10.3390/plants10010006

Author Response

Chile, march 12/ 2021

Professor Dr. Reviewer 1

Dear Reviewer 1,

We are pleased to inform you that we are submitting the revised version of the manuscript nanomaterials-1641626: “SYNTHESIS OF BIOI/MORDENITE COMPOSITES FOR PHOTOCATALYTIC TREATMENT OF ORGANIC POLLUTANTS PRESENT IN AGRO-INDUSTRIAL WASTEWATER”  by A. Gallegos-Alcaíno, N. Robles-Araya, C. Avalos, A. Alfonso Alvarez, C.A. Rodríguez, Héctor Valdés, N.A. Sánchez-Flores, Juan C. Durán-Alvarez, Monserrat Bizarro, Francisco J. Romero-Salguero and Adriana C. Mera,  for possible publication in the Nanomaterials Journal.

We appreciate the time and valuable comments from each reviewer to improve our manuscript. The manuscript was accordingly revised based on the reviewers comments. We hope the modifications made to the document and figures fulfill the requirements of journal and reviewers.

Please let us know if any additional information is required.

Sincerely,

Ph.D Adriana C. Mera

Academic-Researcher

Instituto de Investigación Multidisciplinario en Ciencia y Tecnología (IIMCT).

Universidad de La Serena

La Serena, Chile.

Response to Reviewer 1 Comments

We greatly appreciate the comments and corrections made by reviewer 1 to our manuscript. We have made our best efforts to address all your questions, comments, and suggestions. Herein you will find the point-by-point answers to your comments. 
We appreciate the critical evaluation to improve our manuscript. We hope the modifications made to the document and figures fulfill your requirements.

Reviewer 1

This paper describes the synthesis of BIOI/Mordenite composites and their application to the photocatalytic treatment of organic pollutants present in agro-industrial wastewater. I accept this paper for publication after minor revision based on the following reasons. 1. The paper is written well. 2. All the methods and characterization results are very well explained. 3. The conclusions and findings are well supported by the results. 4. The research design is scientifically aligned

Comment 1. English editing is required.

Response 1: Thank you for the comment. We have subjected the manuscript to an English revision by a language expert. Please see the attachment.

Comment 2. There are typo errors that should be revised.

Response 2: We apologize for these typo errors. We have carried out an in-depth review of the manuscript to delete these errors. Please see the attachment.

Comment 3. The authors should emphasize the importance of BIOI/Mordenite in reference to its efficiency and application for wastewater treatment.

Response 3: Thank you very much for your comment. We have included in the new version of the manuscript the information to emphasize the importance of BiOI/mordenite. Please see the Introduction section (page 4). The modifications are highlighted in yellow on the manuscript. Please see the attachment.

Comment 4. The authors must describe in detail why they selected BIOI/Mordenite for their current research.

Response 4: Thanks for your comment. There are many different kinds of synthetic zeolites; however, modernite-type synthetic zeolite stands out because it has a high ratio of silicon to aluminum atoms; this characteristic makes mordenite more resistant to acids than other zeolites. This information, and more details have been added in the introduction section of the revised version of the manuscript (please see page 4).  In addition, preliminar results obtained by our research groups have shown that BiOI coupled to mordenite exhibited higher photocatalytic efficiency when compared with other kinds of zeolites. However, these results are not shown in the manuscript, because we consider that they are not in the focus of the work, and may confuse the readers. Please see the attachment.

Comment 5. The authors should add a brief explanation or a table describing the comparison between the efficiency of their synthesized material with others already reported in the literature

Response 5: Thank you very much for this suggestion. It is important to mention that comparing the degradation efficiency of caffeic acid by using BiOI/mordenite with other materials, compounds or different experimental conditions can be misleading. This is because there are many different parameters and conditions that affect the degradation efficiency, for instance, the contaminant nature, the chemical interaction between contaminant and the semiconductor surface, the amount of photocatalyst, the power light intensity, the solution pH, point of zero charge, specific surface, surface functional groups and others. Considering this, we have elaborate the requested table, which is shown below (see Table R.1). However, we consider that this information can be confusing in the manuscript. Accordingly, it was not included in the revised version of the manuscript.

Table R.1. Comparison of the photocatalytic efficiency of BiOI/mordenite with other similar materials reported in the literature.

Authors

Composite

Synthesis Method

Target

Compound

Photocatalytic

Efficiency

Reference (DOI)

L. Zhao et al 2014

BiOI/ nature zeolite 

Hydrothermal

Methylene blue

(MB)

94.8%

visible light

60 min

10.1039/C4RA07049F

Q. Whag et al 2017

BiOBr/ H3PW12O14 zeolite

Impregnation

Methyl orange (MO)

72%

visible light

75 min

10.1039/C7NJ00543A

Q. Wang et al 2015

BiOBr/ MCM-41 zeolite

Impregnation

Methyl orange (MO)

75%

visible light

60 min

10.1016/j.matlet.2015.08.111

A. Samadi-Maybodi and S. Massometh Pourali 2013

BiOCl/ zeolite P

Microwave-assisted conventional hydrothermal method

Photocatalytic efficiency is not tested.

The study only characterizes the material obtained.

10.1016/j.micromeso.2012.02.012

G. Zhang et al 2017

BiOCl/TiO2-zeolite

Hydrolysis-precipitation

Rhodamine B

(RhB)

72.4%

visible light

90 min

10.1016/j.jtice.2017.09.030

G. Xiaoya et al 2019

PO4/

HZSM5/ BiOCl

hydrothermal

Carbamazepina (CBZ)

83.2%

solar light irradiation

30 min

10.1016/j.jtice.2019.08.022

H. Salari et al 2016

BiOCl/

BiVO4/

mordenite

hydrothermal

Acid blue 92 (AB92)

visible irradiation

10.1080/14328917.2016.1264849

Comment 6. The authors should cite some important work related to the degradation of pollutants.

https://doi.org/10.1016/j.jenvman.2021.112605

https://doi.org/10.3390/plants10010006

Response 6: Thank you for the suggestions. We have included the mentioned references in the Introduction section (please see page 3) as reference [9] and [10]. Please see the attachment.

Reviewer 2 Report

In this manuscript, the authors focused on the development of BiOI/mordenite composite and used in photocatalytic degradation of caffeic acid. The topic of the manuscript can be considered as interesting for the readers. The manuscript is generally well written with a logic structure.  Hence, I recommended a major revision and it is accepted for this journal after the author clarifies the following comments.

  1. What is the reason to study the photocatalytic degradation of caffeic acid? Is this compound more abundant in wastewater as compared to other pollutants of emerging concern? Is it more toxic?
  2. Section 2.6: In specific surface area analysis, which pressure and temperature were used for degassing procedure? Which P/P0 values were used in the calculation of specific surface areas?
  3. Section 3.2: The legend of figure 6 is missing.
  4. The authors should make comparison with literature for the pollutant used.
  5. I Suggest the authors to discuss briefly the scale-up of the process (problems, possibilities), and the economy and/or energy efficiency of the process (related to UV generation, for instance).
  6. Which separation methods are suggested for the catalysts recycling (for the practice)?

Author Response

Chile, march 12/ 2021

Professor Dr. Reviewer 2.

Dear Reviewer 2,

We are pleased to inform you that we are submitting the revised version of the manuscript nanomaterials-1641626: “SYNTHESIS OF BIOI/MORDENITE COMPOSITES FOR PHOTOCATALYTIC TREATMENT OF ORGANIC POLLUTANTS PRESENT IN AGRO-INDUSTRIAL WASTEWATER”  by A. Gallegos-Alcaíno, N. Robles-Araya, C. Avalos, A. Alfonso Alvarez, C.A. Rodríguez, Héctor Valdés, N.A. Sánchez-Flores, Juan C. Durán-Alvarez, Monserrat Bizarro, Francisco J. Romero-Salguero and Adriana C. Mera,  for possible publication in the Nanomaterials Journal.

We appreciate the time and valuable comments from each reviewer to improve our manuscript. The manuscript was accordingly revised based on the reviewer’s comments. We hope the modifications made to the document and figures fulfill the requirements of journal and reviewers.

Please let us know if any additional information is required.

Sincerely,

Ph.D Adriana C. Mera

Academic-Researcher

Instituto de Investigación Multidisciplinario en Ciencia y Tecnología (IIMCT).

Universidad de La Serena

La Serena, Chile.

Response to Reviewer 2 Comments.

We greatly appreciate the comments and corrections made by reviewers to our manuscript. We have made our best efforts to address all your questions, comments, and suggestions. Herein you will find the point-by-point answers to your comments.

We appreciate the critical evaluation to improve our manuscript. We hope the modifications made to the document and figures fulfill your requirements.

Reviewer 2

In this manuscript, the authors focused on the development of BiOI/mordenite composite and used in photocatalytic degradation of caffeic acid. The topic of the manuscript can be considered as interesting for the readers. The manuscript is generally well written with a logic structure.  Hence, I recommended a major revision and it is accepted for this journal after the author clarifies the following comments.

Comment 1. What is the reason to study the photocatalytic degradation of caffeic acid? Is this compound more abundant in wastewater as compared to other pollutants of emerging concern? Is it more toxic?

Response 1: Thank you very much for your comment. We have included in the new version of the manuscript the reason for studying the degradation of caffeic acid. Please see the Introduction section (page 3) Please see the attachment. Modifications in this section are highlighted with yellow color in the manuscript. More information about the choice of caffeic acid is given below, but not included in the manuscrit due to brevity.

Wine production is one of the most important agricultural industries for countries like Italy, Spain, Australia, Brazil, Chile, China, France, Germany, India, South Africa and the United States of America [1, 2]. In Chile, an important industry is wine and pisco production. It is estimated that in the wine industry, 75% of total waste corresponds to wastewater. Usually from 2 to 14 L of residual water is produced for each liter of bottled wine, and in the best of cases, it generates one liter of wastewater for every liter of wine produced [3].

Although, at a global level, wine residues present important variations in quantity and characteristics, these effluents are characterized by high concentrations of organic compounds, from which 85% is soluble and consisting mainly of organic acids, alcohols (ethanol and methanol), sugars (e.g. glucose, fructose), proteins and polyphenols. The latter gives a remarkable refractory character [4-5].

Two groups of phenolic compounds are present in these wastewaters, approximately 30% to 80% correspond to simple phenolic compounds (p-coumaric acid, gallic acid, catechol, gentisic acid, caffeic acid). The second group corresponds to complex phenolic compounds (tannins and anthocyanins) that represent 8% to 10% of the total [6-8].

Some research shows that phenolic compounds, commonly present in waste from the wine industry, are toxic to microorganisms and plants. They are known to be harmful to fish and are suspected of being mutagenic and carcinogenic. For example, the discharge of water from bodega, between 1930 to 1940, gave rise to the migration of salmon populations to northern California [9].

It is necessary to highlight that phenolic compounds are responsible for the inhibitory effects on microbial activity in biological treatment systems, which have been used for the remediation of wine residues.

Currently, many of the effluents that receive treatment in the industries dedicated to the production of wine and pisco in Chile do not comply with Chilean environmental regulations to be used in irrigation or to be discharged into bodies of water [10-13].

Thus, the adoption of advanced technical tools (as heterogeneous photocatalysis) of pretreatment gain importance that allow increasing its competitive capacity in the face of the growing demands and standards of the international market acquires relevance for this industry.

References

[1] O.i.v.a.w.I. organisation, Statistics of the world vitiviniculture sector, 2011

[2] R.R.Ganesh, J. V. Thanikal, R. A. Ramanujam,M. Torrijos, Anaerobic treatment of winery wastewater in fixed bed reactors, Bioprocess Biosyst Engineery, 33 (2010) 619–628.

[3] A. Antonio J. Pirra, L. Arroja, I. Capela, The influence of port wine in winery effluent production Fresenius Environmental Bulletin, 19 (2010) 3177-3184.

[4] A.L. Eusebi, G. Gatti, P. Battistoni and F. Cecchi, from conventional activated sludge to alternate oxic/anoxic process: the optimisation of winery wastewater treatment, Water Science & Technology, 60 (2009) 267-287.

[5] A.D. Marco S. Lucas, Rui M. Bezerra, Jose A. Peres, Gallic acid photochemical oxidation as a model compound of winery wastewaters, Environmental Science and Health Part A 43 (2008) 1288–1295.

[6] A. D. Bravo, V. Valdivia,M. Torrijos, G. Ruiz-Filippi, R. Chamy, Anaerobic sequencing batch reactor as an alternative for the biological treatment of wine distillery effluents, Water Science & Technology, 60 (2009) 1155-1160.

[7] A.B. Casares Faulín, Análisis de polifenoles en los vinos mediante técnicas de separación, Química (Ed.), Universidad Politécnica de Catalunya (UPC) Barcelona, España, 2010.

[8] Z. Piñeiro Méndez, E. López Cabarcos, Desarrollo de nuevos métodos de extracción para el análisis de compuestos de interés enológico, Cádiz (Ed.), Universidad de Cádiz, Cádiz, España, 2005. 191

[9] G.T. Heather, L. Shepherd, Treatment of High-Strength Winery Wastewater using a Subsurface Flow Constructed Wetland Water, environmental research, 73 (2001) 394-404.

[10] F.C. Quiroz, Tratamiento de RILes vitivinícolas, Trabajo de titulación para obtener el título de ingeniero ambiental Universidad Santiago de Chile, Santiago de Chile, 2007, pp. 1 -138.

[11] Chile que produce limpio, Acuerdo de Producción más limpia de la industria Vitivinícola Chilena, in: G.d. Chile (Ed.), Consejo Nacional de Producción Limpia, Santiago de Chile, 2003.

[12] Tecnolimpia, Guia de manejo sustentable de RILes vitivinícolas, in: C.N.d.P.m. Limpia (Ed.), Gobierno de Chile, Santiago de Chile, 2010.

[13] Manejo de Herramientas en Producción Limpia, Medio ambiente y producción más limpia, in: C.N.d.P. Limpia (Ed.), Gobierno de Chile, Santiago de Chile, 2011.

 Comment 2. Section 2.6: In specific surface area analysis, which pressure and temperature were used for the degassing procedure? Which P/P0 values were used in the calculation of specific surface areas?

Response 2: Thank you for your comment. The degassing procedure is carried out in a Smart VacPrep Micromeritics equipment. The pressure and temperature used were 10-2 - 10-3 mmHg and 150 ºC (by 3h), respectively. The used values for specific surface calculation were performed by the Rouquerol function and the P/P° range for all these materials was 0.05-0.3. The single point surface area used was P/P° = 0.024.

 Comment 3. Section 3.2: The legend of figure 6 is missing.

Response 3: Thank you. The legend of Fig 6 is already included in the manuscript. Please find the following legend in page 16 (Please see the attachment):

Figure 6. X-ray diffraction patterns of synthesized materials: (a) pure BiOI, (b) mordenite, (c) BiOI/mordenite composite synthesized at 153°C during 20.3 hours, (d) BiOI/mordenite composite obtained at 180°C during 7 hours, (e) optimized BiOI/mordenite composite synthesized at 187°C during 9 hours.

Comment 4. The authors should make comparisons with literature for the pollutant used.

Response 4: Thank you for your kind suggestion. We elaborated Table R.2 (see below) comparing different advanced oxidation processes for the degradation of caffeic acid (CA). However, the objective of our work is the optimization of the BiOI/Modernite 90 synthesis for contaminants degradation. In this work we selected CA as the target molecule for determining the composite with the highest photocatalytic efficiency. Compare the degradation efficiency of caffeic acid by using other materials or different experimental conditions can be confusing. This is because there are many different parameters and conditions that affect the degradation efficiency of a pollutant, such as the semiconductor material or composite, surface characteristics, the amount of photocatalyst, the power light intensity, the solution pH, point of zero charge, specific surface, surface functional groups and others. In addition, the characterization techniques to study the degradation of pollutants are different (e.g. UV- Vis spectrophotometry, HPLC and others), and it makes the comparison between works complicated. Accordingly, Table R.2 was not included in the revised version of the manuscript.

Table R.2. Comparison of the efficiency with literature for the pollutant used.

Authors           

Material

Efficiency (%)

Conditions

Reference (DOI)

Vendetti et al 2015

Carbon doped TiO2

92, 75, 65, 48

Different concentration of AC (18, 36, 54, 90) mgL-1

10.1021/acs.langmuir.5b00560

García-Montelongo et al 2015

TiO2

90

pH=5.2,  dosing 1.1 g/L. 30 min under lamp irradiation

10.2166/wst.2015.039

Ren et al 2021

DBD plasma and Ce1Co9OOH/ DBD Catalyst

75.6 at 0.8 g/kWh,     100 at 2.03 g/kWh

Plasma and catalysis. High energy consumption

10.1016/j.jhazmat.2020.123772

Diaz et al 2021

BiOBr

60, 77

Microspheres synthesized using the solvothermal method.

10.1016/j.mseb.2021.115432

Puga et al 2022a

ZnBiO-SG-500

ZnBiO-HT-500

52.7 (UV), 51.1 (Visible)

39.2 (UV),  61.4 (Visible)

Sol-gel and hydrothermal process

10.1016/j.apsusc.2021.152351

Yañez et al 2016

TiO2 P-25

80

 pH 5.3 and 0.9 g L-1 of TiO2. The photocatalytic degradation was monitored by HPLC under UV-A light.

10.1080/10934529.2015.1086211

Sponza &Oztekin 2015

Magnetic nickel–

coated carbonbased TiO2

88

Sintered under N

2(g) atmosphere at

400 °C for 2 h

10.1155/2015/276768

Baransi et al 2012

TiO2-PAC

87

The photocatalytic degradation was performed with a suspended mixture of TiO2 and powdered activated carbon (PAC) at pH = 3.4 and 8, under different UV light

10.1016/j.watres.2011.11.049

Puga et al 2021

AgBr/SnO2

85

Hydrothermal followed by precipitation-deposition

10.1016/j.seppur.2020.117948

Puga et al 2022b

{001} faceted TiO2

sensitized with

AgBr or Ag3PO4

75

Hydrothermal followed by precipitation-deposition

10.1016/j.mseb.2021.115555

Comment 5. I Suggest the authors to discuss briefly the scale-up of the process (problems, possibilities), and the economy and/or energy efficiency of the process (related to UV generation, for instance).

Response 5: Thank you very much for your suggestion. We agree that the scale-up of the process, economy and energy efficiency are interesting issues to address. However, it has to be noted that all these aspects are out of the scope of this research. Before studying and discussing the scale-up of the process, there are many other fundamental aspects to solve, such as: optimization of solution pH, amount of photocatalyst, caffeic acid concentration, recovery tests, and others. Particularly, this work is focused in the optimization of experimental conditions during the synthesis of BiOI/mordenite composites for the higher degradation efficiency of caffeic acid.  By the way, in order to address your suggestion, we have added some comments about the advantage of using BiOI in terms of improving the energy efficiency of the process, please see the Introduction section (page 3). Please see the attachment.

Comment 6. Which separation methods are suggested for the catalysts recycling (for the practice)?

Response 6: Thank you very much for your comment. Our work is still in an initial stage of investigation. Dimensioning on an industrial scale, requires delving into more aspects and carrying out more tests. However, to address the reviewer question, we consider that one of the potential methods for the practical separation of this composite is the decantation because the support of the semiconductor (zeolite) is denser when compared with single semiconductor.

Round 2

Reviewer 2 Report

Authors have revised the manuscript according the recommendations, and answered the questioned points. Now it looks suitable for publication in Nanomaterials.